# LINK PREDICTION USING NEUMANN EIGENVALUES

## ABSTRACT

Link prediction (LP) is a crucial task in graph-structured data, aiming to estimate the likelihood of non-observable links based on known graph structure and node/edge features. Despite the success of Graph Neural Networks (GNNs) in solving graph-level tasks, their results, compared to classical methods, are worse in solving node-level tasks such as LP. The main reason lies in the limitations of Message Passing GNNs (MPNNs), the most common technique used in GNNs. One of the main limitations of MPNNs is their inability to distinguish between some graphs, e.g., k-regular graphs. Discriminating between k-regular graphs lets us count the sub-structures and triangles, which are crucial in the success of classical methods for the LP task. Encoding link representation instead of node representation can solve this problem, but the previous methods are prohibitively expensive and thus impractical. We propose a novel light learnable eigenbasis to encode the link representation and induced subgraphs (sub-structures) efficiently and explicitly. We propose formulating the linear constraints as the eigenvalue problem with linear constraints. We efficiently implement our proposed convolutional layer with a novel *learnable Lanczos algorithm with linear constraints, LLwLC*. Specifically, we introduce Neumann eigenvalues and encode its corresponding constraints to the eigenbasis. Given the Neumann constraints, the Neumann basis splits the nodes into two (one-hop and two-hop away nodes) and efficiently encodes the relation between them. We also investigate the effect of encoding different linear constraints (subgraphs). Although our theoretical results apply to many problem settings, we report our results on link prediction tasks achieving state-of-the-art in benchmark datasets.

## 1 INTRODUCTION

We observe the ubiquitous existence of graphs in different applications such as social networks Adamic & Adar (2003), citation networks Shibata et al. (2012), knowledge graph construction Nickel et al. (2015), metabolic network reconstruction Oyetunde et al. (2017), and recommender systems (Monti et al. (2017); Nickel et al. (2014)). Recent state-of-the-art methods to process graph-structured data are based on graph neural networks (GNNs) Kipf & Welling (2016). Spatial GNNs primarily utilize the MPNN Gilmer et al. (2017) frameworks mainly because of their simplicity and scalability. However, their performance is worse than classical methods in node-level tasks (e.g., LP) due to their limited *expressive power*. MPNNs, based on neighborhood aggregation schemes, are at most as expressive as classical Weisfeiler-Lehman (1-WL Weisfeiler & Leman (1968)) test Xu et al. (2019); Morris et al. (2019) thus not capable of discriminating between some graphs, e.g., k-regular graphs. Chen et al. (2020) proved that MPNNs cannot count connected subgraphs with three or more nodes that form cycles. Distinguishing between k-regular graphs and counting the sub-structures and triangles are crucial to the success of classical methods for the LP task Chamberlain et al. (2022). Specifically, the subtrees MPNN build are the same for both subgraphs in Figure 1.

One major reason for inexpressiveness in MPNNs is their incapability to count cycles. To address this issue, the solutions fall into three main directions: Aligning to the k-WL hierarchy Maron et al. (2019b); Keriven & Peyré (2019); Azizian & Lelarge (2021), augmenting node features with identifiers, or utilizing the structural information that cannot be captured by the WL test Bodnar et al. (2021b;a). Our proposed solution falls into the last category.

Subgraph GNNs (SGNNs) are state-of-the-art for solving expressivity in MPNNs. SGNNs Bevilacqua et al. (2022); Guerra et al. (2022) enrich GNN features by encoding extracted SGNNs as new features

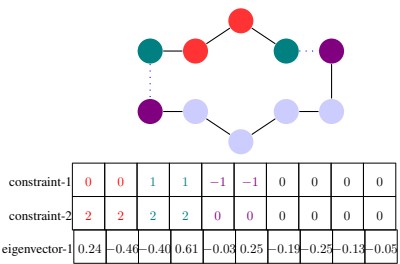 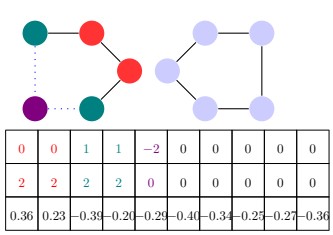

| constraint-1 | 0 | 0 | 1 | 1 | −1 | −1 | 0 | 0 | 0 | 0 |
|---|---|---|---|---|---|---|---|---|---|---|
| constraint-2 | 2 | 2 | 2 | 2 | 0 | 0 | 0 | 0 | 0 | 0 |
| eigenvector-1 | 0.24 | −0.46 | −0.40 | 0.61 | −0.03 | 0.25 | −0.19 | −0.25 | −0.13 | −0.05 |

| | 0 | 0 | 1 | 1 | −2 | 0 | 0 | 0 | 0 | 0 |
|---|---|---|---|---|---|---|---|---|---|---|
| | 2 | 2 | 2 | 2 | 0 | 0 | 0 | 0 | 0 | 0 |
| | 0.36 | 0.23 | −0.39 | −0.20 | −0.29 | −0.40 | −0.34 | −0.25 | −0.27 | −0.36 |

Figure 1: Two k-regular subgraphs with their corresponding Neumann constraints (constraint-1 and constraint-2) and LLwLC computed Neumann eigenvector (eigenvector-1). For the two k-regular subgraphs, the MPNN makes the same tree, while our proposed eigenbasis (*LLwLC*) can distinguish between them. The red, teal, and blue nodes are the query, one-hop-away, and two-hop-away nodes. The sum of the dotted edges (first Neumann constraint: $\sum_{x \sim y} (\mathbf{f}(x) - \mathbf{f}(y) = 0)$ is zero.

and passing them to the GNN architecture. Previous works did the pre-processing steps to encode the information about the set of nodes and lifting graphs into simplicial or cell complexes Bodnar et al. (2021b;a)). However, they require an expensive pre-computation stage in the worst-case scenario. Also, Bevilacqua et al. (2022) considered a set of nodes a new set of features. They equivariantly encoded the subgraphs extracted with the fixed policy and injected it into the GNN architecture as a new set of features. Despite their success, they are prohibitively expensive and exhaustive.

We propose a novel method to encode the subgraphs into our proposed GNN model. Our method stems from graph signal processing (GSP) Ortega et al. (2018) and spectral graph theory Chung & Graham (1997). We introduce a new eigenbasis, *Learnable Lanczos Layer with Linear Constraints (LLwLC)*, that encodes the linear constraints, mainly the extracted subgraphs, into the basis. We devise our proposed low-rank approximation Eckart & Young (1936) of the Laplacian matrix based on the Lanczos algorithm with linear constraints Golub et al. (2000).

The new basis makes the features more expressive by explicitly encoding the linear constraints over the graph. We investigate linear constraints with different subgraph extraction policies. We mainly conduct our experiments with the vertex-deleted subgraphs and Neumann eigenvalue constraints Chung & Graham (1997). The former is beneficial to show that LLwLC is a universal approximator (specifically with a few linear constraints). The latter is beneficial by encoding the boundary conditions of the input graph as new linear constraints into the eigenbasis (link representation between one hop and two hop away nodes leads to counting the triangle and substructures). The Neumann constraints let GNNs distinguish between the k-regular graphs from each other, as shown in Figure 1.

Theoretically, LLwLC can be conducted in many problem settings. We analyze its effectiveness on challenging LP tasks where addressing k-regular graphs and counting the substructures are crucial for their success. SEAL Zhang & Chen (2018) and BUDDY Chamberlain et al. (2022) considered subgraphs to encode information for the LP task. However, neither SEAL nor BUDDY, which only leverages oversimplified pairwise node representation features, have enough expressive capability to distinguish between input graphs.

The **main contributions** of our paper are (i) Formulating an eigenvalue problem with linear constraints utilizing the Lanczos algorithm with the linear constraints and proposing a novel light learnable eigenbasis to encode sets of linear constraints (induced subgraphs) into the basis *(LLwLC)*. (ii) Proposing and investigating the Neumann eigenvalue constraints in our new eigenbasis (not only it encodes the induced subgraphs but also the link representation). (iii) We provide a rigorous theoretical analysis of our proposed eigenbasis *LLwLC*. (iv) We show our method is a universal approximator (as expressive as the k-WL test) with linear order of complexity w.r.t input graph nodes.

## 2 PRELIMINARIES.

**Notations.** $G(V, E, \mathbf{X})$ is a graph with vertex set $V$, edge set $E$, and *node features* $\mathbf{X} \in \mathbb{R}^{|V| \times d}$. Each column of $\mathbf{X} : \mathbf{x}_v \in \mathbb{R}^d$ refers to the features on the node $v \in V$. $\mathbf{A}$ and $\mathbf{D}$ are the graph

adjacency and graph degree matrices on $G$, respectively. $\mathbf{L} = \mathbf{D} - \mathbf{A}$ is the graph Laplacian on $G$. $n$ is the number of nodes. $d_x$ is the degree of node $x$ in $G$.

**Lanczos Algorithm with Linear Constraints.** For a given symmetric matrix $\mathbf{L} \in \mathbb{R}^{n \times n}$ and a randomly initialized vector $\nu \in \mathbb{R}^n$, the N-step Lanczos algorithm computes an orthogonal matrix $\mathbf{Q} \in \mathbb{R}^{n \times m}$ and a symmetric tridiagonal matrix $\mathbf{T} \in \mathbb{R}^{m \times m}$, such that $\mathbf{Q}^\top \mathbf{L} \mathbf{Q} = \mathbf{T}$. We denote $\mathbf{Q} = [\mathbf{q}_1, \ldots, \mathbf{q}_N]$ where column vector $\mathbf{q}_i$ is the $i$-th Lanczos vector. $\mathbf{T}$ is the tridiagonal matrix with the eigenvector and eigenvalue matrices $\mathbf{B} \in \mathbb{R}^{m \times m}$ and $\mathbf{R} \in \mathbb{R}^{m \times m}$, respectively. $\mathbf{Q}$ forms an orthonormal basis of the Krylov subspace $\mathcal{K}_N(\mathbf{L}, \mathbf{b})$ and its first K columns form the orthonormal basis of $\mathcal{K}(\mathbf{L}, \mathbf{x})$. By investigating the $j$-th column of the system $\mathbf{L}\mathbf{Q} = \mathbf{Q}\mathbf{T}$ and rearranging terms, we obtain $\mathbf{L}\mathbf{q}_j = \beta_{j+1}\mathbf{q}_{j+1} + \beta_j \mathbf{q}_{j-1} + \alpha_j \mathbf{q}_j$, and the first $j$ steps of the Lanczos process take the form $\mathbf{L}\mathbf{Q}_j = \mathbf{Q}_j \mathbf{T}_j + \beta_{j+1}\mathbf{q}_{j+1}\mathbf{e}_j^\top$ Liao et al. (2019). Having the linear constraint changes the plain Lanczos algorithm (Algorithm 1) by replacing $\mathbf{u_j} = \mathbf{L}\mathbf{q}_j - \beta_j \mathbf{q}_{j-1}$ with $\mathbf{u}_j = \mathbf{p}_j - \beta_j \mathbf{q}_{j-1}$ assuming the initial vector $\nu$ is projected into the null space of the constraints ($\nu \in \mathcal{N}(\mathbf{C}^\top)$) Golub et al. (2000). If we project the initial vector $\nu$ into null space of the constraint matrix $\nu_1 = \mathbf{P}\nu \in \mathcal{N}(\mathbf{C}^\top)$ and notice the mathematical equivalence between computing the smallest eigenvalue of the constraint $A_p = \mathbf{P}^\top \mathbf{L} \mathbf{P}$ and $\mathbf{L}$ then one step of the Lanczos algorithm with the linear constraints is $\beta_{j+1}\mathbf{q}_{j+1} = \mathbf{P}\mathbf{L}\mathbf{P}\mathbf{q}_j - \beta_j \mathbf{P}\mathbf{q}_{j-1} - \alpha_j \mathbf{P}\mathbf{q}_j = \mathbf{P}(\mathbf{L}\mathbf{q}_j - \beta_j \mathbf{q}_{j-1} - \alpha_j \mathbf{q}_j)$. This means projecting to the null space of the constraint matrix in each step of the algorithm.

**Proposition.** If we start simple Lanczos with $\nu \in \mathcal{N}(\mathbf{C}^\top)$, then $\mathbf{q}_j \in \mathcal{N}(\mathbf{C}^\top)$ for all $j$.

**Spectral Graph Convolutional Networks.** The early graph learning models rooted in graph signal processing (GSP) Ortega et al. (2018), trying to generalize signal processing convolution operators on graphs. Among definitions of frequency representations of graph signals Ortega et al. (2018), which are grounded on spectral graph theory Chung & Graham (1997) and graph wavelet theory Hammond et al. (2011), the spectral graph theory-based one is most popular. It defines the graph Fourier transform and its inverse based on the eigenbasis of the graph Laplacian. The graph Laplacian $\mathbf{L}$ is positive semi-definite and can be factored as $\mathbf{L} = \mathbf{U}\boldsymbol{\Lambda}\mathbf{U}^\top$, where $\mathbf{U} = [\mathbf{u_1}, \ldots, \mathbf{u_n}] \in \mathbb{R}^{n \times n}$ denotes the matrix of eigenvectors, sorted according to their eigenvalues. Further, the matrix $\boldsymbol{\Lambda}$ is a diagonal matrix with $\Lambda_{i,i}$ is a $\lambda_i$, where $\lambda_i$ denotes the $i^{th}$ eigenvalue. For a graph signal $\mathbf{x} \in \mathbb{R}^n$, the graph Fourier transform Shuman et al. (2013) and its inverse are $\mathbf{U}^\top \mathbf{x}$ and $\mathbf{U}\mathbf{x}$, respectively. Hence, the graph Fourier transform is an orthonormal (linear) transform to the space spanned by the basis of the eigenvectors in $\mathbf{U}$. Based on this observation, spectrum-based methods generalize convolution to graphs. Hence, the graph convolution is $\mathbf{U}(\mathbf{U}^\top \mathbf{x} * \mathbf{U}^\top \mathbf{y}) = \mathbf{U}g(\Lambda)\mathbf{U}^\top \mathbf{x}$, where $\mathbf{y}$ is the graph filter, $g$ is the function applied over the eigenvalue matrix $\boldsymbol{\Lambda}$ to encode the graph filter, and $*$ is the elementwise multiplication. The seminal spectral GCN method Bruna et al. (2013) defined $g(\Lambda)$ to be $\Theta_{i,j}$. However, the eigendecomposition is cubic in the number of nodes. To address this, different $g$ functions were defined Henaff et al. (2015); Kipf & Welling (2016). Hammond et al. (2011) estimated the graph filter with the Chebychev function. Kipf & Welling (2016) made the Chebyshev Spectral GCNs more scalable by setting $\mathbf{x} * g_\theta \approx \Theta(\mathbf{I}_n + \mathbf{D}^{-\frac{1}{2}}\mathbf{A}\mathbf{D}^{-\frac{1}{2}}\mathbf{x})$. The drawbacks of Kipf & Welling (2016) are that first, it only propagates information from any node to its nearest neighbours, *i.e.*, nodes that are one-hop away, and second, no learnable parameters are associated with the graph Laplacian $\mathbf{L}$. The only learnable parameter is a linear transform applied to every node simultaneously. There are only two ways to consider k-hops away nodes with ChebychevNet Kipf & Welling (2016): either stacking layers on top of each other (which leads to over-smoothing Li et al. (2018)), or computing the powers of $\mathbf{L}$, which is computationally expensive. Instead, LanczosNet Liao et al. (2019) utilizes the Lanczos algorithm and proposes a spectral GCN model to leverage multi-scale information via fast-approximated computation of matrix power and have learnable spectral filters. Like previous spectral and spatial GCNs, LanczosNet does not encode the relation between sets of nodes (subgraphs) and link representation. Thus, their features have limited expressivity, which leads to, e.g., not being able to distinguish between the k-regular graph.

To address the above issues and make features more expressive, our approach advocates using a novel learnable spectral basis encoding the subgraphs (as linear constraints). Specifically, we investigate the Neumann eigenvalue constraints as a new basis for our proposed GNN architecture. We formulate it as an eigenvalue problem with linear constraints utilizing the Lanczos algorithm with the linear constraints Golub et al. (2000) to solve it. In the next section, we describe our method in detail.

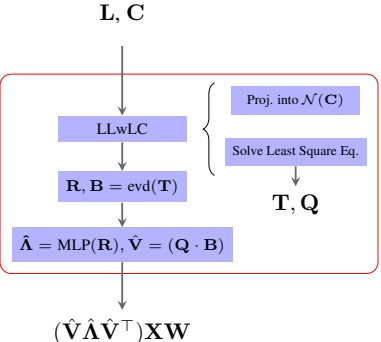

**L, C**

LLwLC

Proj. into $\mathcal{N}(\mathbf{C})$

Solve Least Square Eq.

$\mathbf{R}, \mathbf{B} = \text{evd}(\mathbf{T})$

$\hat{\mathbf{\Lambda}} = \text{MLP}(\mathbf{R}), \hat{\mathbf{V}} = (\mathbf{Q} \cdot \mathbf{B})$

**T, Q**

$(\hat{\mathbf{V}}\hat{\mathbf{\Lambda}}\hat{\mathbf{V}}^\top)\mathbf{XW}$

Figure 2: In each block, the low-rank approximation of the graph Laplacian matrix (**L**) is estimated with the constraints (**C**) encoded in it. To compute the Lanczos algorithm with the linear constraints, in each step, we project into the null space of the constraint matrix ($\mathcal{N}(\mathbf{C}^\top)$) and solve the least square equation with the QR method. **T** is the tridiagonal matrix which its eigendecomposition returns eigenvalue matrix (**R**) and eigenvector matrix (**B**). Finally, by applying multi-layer perceptron (MLP) over **R** and reconstructing **V**, we have LLwLC eigenbasis. **X** and **W** are feature and weight matrix, resepectively.

---

**Algorithm 1** LLwLC. Differences between the Lanczos-Net and LLwLC are shown in red.

---

**input** : $\mathbf{L}, \mathbf{P}, \nu, \kappa, \epsilon$
**init** : $\nu_1 = \mathbf{P}(\nu), \beta_1 = \|\nu_1\|_2, q_0 = 0$.
For $j = 1$ to $\kappa$:

$\quad \mathbf{q}_j = \frac{\nu_j}{\beta_j}$

$\quad \mathbf{u_j} = \mathbf{L}\mathbf{q}_j - \beta_j\mathbf{q}_{j-1}$

$\quad \alpha_j = \mathbf{u}_j^\top\mathbf{q}_j$

$\quad \color{red}{\mathbf{p}_j = \mathbf{P}(\mathbf{L}\mathbf{q}_j) \rightarrow \text{LSQR}}$

$\quad \color{red}{\mathbf{u}_j = \mathbf{p}_j - \beta_j\mathbf{q}_{j-1}}$

$\quad \nu_{j+1} = \mathbf{u}_j - \alpha_j\mathbf{q}_j$

$\quad \beta_{j+1} = \|\nu_{j+1}\|_2$

$\quad$ If $\beta_{j+1} \leq \epsilon$, quit

$Q = [q_1, \ldots, q_\kappa]$
Construct **T**
Eigen decomposition $\mathbf{T} = \mathbf{B}\mathbf{R}\mathbf{B}^\top$
Return $\mathbf{V} = \mathbf{Q} \cdot \mathbf{B}$ and $f(\mathbf{R})$

---

# 3 METHOD

We introduce the novel eigenbasis **learnable Lanczos algorithm with the linear constraints (LLwLC)**. It encodes the relation between a set of nodes/edges by projecting the eigenbasis to the null space of the constraints matrix. LLwLC is a low-rank approximator that simultaneously encodes the constraints (relation between a set of nodes) into the eigenbasis and learns the filters based on the new eigenbasis, which leads to more expressive features.

In the following, first, we explain how we compute the LLwLC eigenbasis, full block, and the full pipeline architecture. Then, we explain the subgraph extraction policies. Finally, we prove the convergence of LLwLC, the novel low-rank approximation eigenbasis.

## 3.1 LLwLC

To devise our proposed eigenbasis, we have to solve a large sparse symmetric eigenvalue problem with homogeneous linear constraints Golub et al. (2000), where $\mathbf{C} \in \mathbb{R}^{n \times l}$ with $n \gg l$ is the constraint matrix:

$$\min_{\mathbf{C}^\top\mathbf{x}=0, \mathbf{x}\neq 0} \frac{\mathbf{x}^\top\mathbf{L}\mathbf{x}}{\mathbf{x}^\top\mathbf{x}}. \tag{1}$$

As discussed in Golub et al. (2000), Equation (1) is mathematically equivalent to computing the smallest eigenvalue of the constraint $\mathbf{P}^\top\mathbf{L}\mathbf{P}$ of matrix $\mathbf{L}$ on the null space $\mathcal{N}(\mathbf{C}^\top)$ of $\mathbf{C}^\top$, where $\mathbf{P}$ is the orthogonal projector onto $\mathcal{N}(\mathbf{C}^\top)$. To solve Equation (1), Golub et al. (2000) derived an inner-outer Lanczos process in Algorithm 1. The algorithm breaks into two parts: the outer and the inner loop, where the outer one is the loop in the simple Lanczos algorithm and the inner one is the loop to solve the least square problem of $\mathbf{P}(\mathbf{b}) = \min_{y\in\mathbb{R}^l}\|\mathbf{C}y - \mathbf{b}\|_2$, where $\mathbf{y} = \mathbf{C}^\dagger\mathbf{b}$ and $\mathbf{b} = \hat{\mathbf{L}}\mathbf{q}_j$.

**Orthogonal Projector.** As mentioned by Golub et al. (2000), if the constraint matrix, **C**, is dense, then **P** is produced by computing the QR decomposition of **C**. For the sparse case, if $\dim(\mathcal{N}(\mathbf{C}^\top)) \approx n$, $\mathbf{P} = \mathbf{I} - \mathbf{C}\mathbf{C}^\dagger$, where $\mathbf{C}^\dagger$ is the Moore-Penrose inverse of **C**. Because of the full column rank assumption on **C**, we have $\mathbf{C}^\dagger = (\mathbf{C}^\top\mathbf{C})^{-1}\mathbf{C}^\top$ ( Björck (1996); Stewart & Sun (1990)).

In contrast to the iterative approach of Golub et al. (2000) to solve the least square equation

$$\mathbf{C}y = \mathbf{b}, \tag{2}$$

given that our constraint matrix is sparse and not large, we utilize the PyTorch framework, which internally uses the direct method of QR factorization Anderson et al. (1992). Thus, it is numerically stable, and we can backpropagate through it.

**Full Block.** We build our complete block, shown in Figure 2. By computing the eigenbasis using the proposed algorithm 1, we learn the features in each block with

$$\sigma(\mathbf{V}f(\mathbf{R})\mathbf{V}^\top\mathbf{X}\mathbf{W}) = \sigma(\hat{\mathbf{L}}\mathbf{X}\mathbf{W}), \tag{3}$$

where $\mathbf{V} = \mathbf{Q} \cdot \mathbf{B}$. We define $f$ to be multi-layer perceptrons (MLPs) over the Ritz values of each block (diagonal matrix $\mathbf{R}$). With the learned filters, we reconstruct our new basis and transform the graph signals, $\mathbf{X} \in \mathbb{R}^{m \times n}$, to this new basis to learn the features that satisfy our constraints. $\mathbf{W} \in \mathbb{R}^{n \times m}$ is the learnable weight. $\sigma$ is the non-linearity we apply in each block where we set it to LeakyReLU in our experiments.

**Full Architecture.** We build our complete pipeline by stacking the LLwLC blocks Figure 2 explained above, followed by a global sort pooling Zhang et al. (2018) and a fully connected block in the last layer. The final output is one value ($y$) corresponding to the existence of the edge. We increase the number of blocks in our pipeline by reusing the eigenbasis we computed once and applying an MLP layer on top of the Ritz eigenvalues.

MPNNs also lose information about the distance between multiple nodes. To make MPNN approaches more expressive, we, same as SEAL Zhang & Chen (2018) and BUDDY Chamberlain et al. (2022), use DRNL, a deterministic instance attribute. As we explained, although LLwLC, in theory, is applicable to different problems, we do experiments in link prediction tasks to show the effectiveness of considering substructures. So, as shown in previous works Zhang & Chen (2018); Chamberlain et al. (2022), considering only two hops away nodes suffices. However, there is no limit to increasing the number of hops because $\mathbf{C}$ only needs to be full column rank.

### 3.2 SUBGRAPH EXTRACTION POLICY.

As defined in ESAN Bevilacqua et al. (2022), the subgraph selection policy is a function $\pi : \mathcal{G} \to \mathbb{P}(\mathcal{G})$, assigned to each graph, where $\mathcal{G}$ be the set of all graphs with $n$ nodes or less and $\mathbb{P}(\mathcal{G})$ be its power set. Although any linear constraint in the input graph satisfying full rank assumption can be encoded in $\mathbf{C}$, we propose the following subgraph extraction policies.

**Neumann Eigenvalue.** The $i^{th}$ Neumann eigenvalue Chung & Graham (1997) is

$$\lambda_{S,i} = \inf_{\mathbf{f}} \sup_{\mathbf{f}' \in C_{i-1}} \frac{\displaystyle\sum_{x \in S} \mathbf{f}(x)\mathbf{L}\mathbf{f}(x)}{\displaystyle\sum_{x \in S} \mathbf{f}^2(x)d_x} \quad \text{s.t.} \quad \sum_{y \in S, y \sim x} (\mathbf{f}(x) - \mathbf{f}(y)) = 0 \quad \sum_{x \in S} \mathbf{f}(x)d_x = 0$$

This can be formulated into

$$\min_{\mathbf{f} \in \mathbb{R}^n} \mathbf{f}^\top \mathbf{L} \mathbf{f} \quad \text{subject to} \quad \|\mathbf{f}\| = 1 \quad \text{and} \quad \mathbf{C}^\top \mathbf{f} = 0, \tag{4}$$

where $\mathbf{L} \in \mathbb{R}^{n \times n}$ is symmetric and large sparse matrix, and $\mathbf{C} \in \mathbb{R}^{n \times l}$ with $n \geq l$ is large sparse and of full column rank. $\mathbf{f} : S \cup \delta S \to \mathbb{R}^n$ is the Neumann eigenvector. The *vertex boundary*, $\delta S$, of an induced subgraph, consists of all vertices that are not in $S$ but adjacent to some vertex in $S$. This is the specific case of an eigenvalue problem with the linear constraints, shown in Equation 1.

**Theorem 1.** *Neumann's features are more expressive than MPNN's features. Besides addressing the node automorphism problem, we can distinguish the k-regular graphs from each other, which leads to more expressivity in GNNs.*

**Constraints C.** The sufficient conditions under which LLwLC can solve graph isomorphism entails that LLwLC is a universal approximator of functions defined on graphs Chen et al. (2019). Given that we can encode any subgraph into our eigenbasis, we can examine whether a specific substructure collection can completely characterise each graph. By the reconstruction conjecture Ulam (1960), we know that we can reconstruct the graph if we have all the $n - 1$ vertex deleted subgraphs.

**Theorem 2.** *If the reconstruction conjecture holds and the substructure collection contains all graphs of size $k = n-1$ with the form of $\displaystyle\sum_{x \in S} \mathbf{f}(x)d_x = 0$ (for every vertex-deleted subgraph S), then LLwLC can distinguish all non-isomorphic graphs of size $n$ and is therefore universal.*

Based on Bollobás (1990), almost every graph has reconstruction number three. Thus, we do not need to extract all the $n - 1$ vertex-deleted subgraphs. This is consistent with our experimental results, where we observed that certain small substructures such as Neumann constraints (or in the other ablation study with only ten vertex-deleted subgraphs) significantly improve the results.

### 3.3 LLwLC CONVERGENCE.

**Perturbation and Error Study.** The accuracy of the linear least square problem using QR factorization depends on the accuracy of the QR factorization. As Zhang et al. (2020) discussed, two types of accuracy error are crucial in QR factorization when solving linear least square problems: The backward error for a matrix $\mathbf{Z}$ is $\frac{\|\mathbf{Z} - \hat{\mathbf{Q}}\hat{\mathbf{R}}\|}{\|\mathbf{Z}\|}$ and the orthogonality of $\hat{\mathbf{Q}}$ is $\|\mathbf{I} - \hat{\mathbf{Q}}^\top \hat{\mathbf{Q}}\|$. Ideally, both numerical errors should be zero, but due to roundoff errors and potential loss of orthogonality of Gram-Schmidt QR, the QR factorization might not be accurate enough to solve the linear least square problem.

The inexact QR factorization to solve Equation 2 will affect the accuracy of both the Lanczos vectors and the tridiagonal matrices generated. So the computed tridiagonal matrix $\mathbf{T}_j$ is a perturbed one of the theoretical tridiagonal matrix, say $\mathbf{T}_j^*$, generated by exact Lanczos iteration is $\mathbf{T}_j = \mathbf{T}_j^* + \mathbf{E}_j$, where $\mathbf{E}_j$ is the perturbation matrix after the $j$-th step. The following theorem represents the error bounds of the computed perturbed tridiagonal matrix compared to the theoretical exact solution of the tridiagonal matrix $\mathbf{T}$ after the $j^{th}$ step of the Lanczos algorithm.

**Theorem 3.** *Let $\mathcal{U}$ and $\tilde{\mathcal{U}}$ be the eigenspaces corresponding to the smallest eigenvalues $\lambda$ and $\tilde{\lambda}$ of the symmetric matrices $\mathbf{L}$ and $\tilde{\mathbf{L}} = \mathbf{L} + \mathbf{E}$, respectively. Then for any $\mathbf{u} \in \mathcal{U}$ and $\tilde{\mathbf{u}} \in \tilde{\mathcal{U}}$ with $\|\mathbf{u}\|_2 = 1$ and $\|\tilde{\mathbf{u}}\|_2 = 1$, we have $\tilde{\lambda} - \lambda \approx \sum_{i=1}^{j} \mathbf{E}_j(i,i)\mathbf{u}(i)^2 + 2\sum_{i=1}^{j-1} \mathbf{E}_j(i, i+1)\mathbf{u}(i)\mathbf{u}(i+1)$, where $\mathbf{E}_j(s,t)$ is the $(s,t)$ element of $\mathbf{E}_j$.*

**Greenbaum's Results Greenbaum (1989).** The tridiagonal matrix $\mathbf{T}_j$ generated at the end of the $j^{th}$ *finite precision* Lanczos process satisfying $\mathbf{L}\mathbf{Q}_j = \mathbf{Q}_j\mathbf{T}_j + \beta_{j+1}\mathbf{q}_{j+1}\mathbf{e}_j^\top + \mathbf{F}_j$, where $\mathbf{e}_j^\top$ is a vector with the $j^{th}$ component one and all the other components zero, $\mathbf{F} = (\mathbf{f}_1, \ldots, \mathbf{f}_j)$ is the perturbation term with $\|\mathbf{f}_j\|_2 \leq \epsilon\|\mathbf{L}\|_2, \epsilon \ll 1$, is the same as that generated by an exact Lanczos process but with a different matrix $\tilde{\mathbf{L}}$. The matrices $\mathbf{L}$ and $\tilde{\mathbf{L}}$ are close in the sense that for any eigenvalue $\lambda(\tilde{\mathbf{L}})$ of $\tilde{\mathbf{L}}$, there is an eigenvalue $\lambda(\mathbf{L})$ of $\mathbf{L}$ such that $|\lambda(\tilde{\mathbf{L}}) - \lambda(\mathbf{L})| \leq \|\mathbf{F}_j\|_2$. Therefore, in our case with the constant accuracy of the QR factorization, we can show $\mathbf{P}\mathbf{L}\mathbf{P}\tilde{\mathbf{Q}}_j = \tilde{\mathbf{Q}}_j\mathbf{T}_j + \beta_j\tilde{\mathbf{q}}_{j+1}\mathbf{e}_j^\top + \tilde{\mathbf{F}}_j$, where $\tilde{\mathbf{F}}_j = \mathbf{O}(\eta)$ with $\eta$ corresponds to the accuracy of the QR method for solving the least square equation.

**Theorem 4.** *Let $\mathbf{U}\mathbf{\Lambda}\mathbf{U}^\top$ be the eigendecomposition of an $n \times n$ symmetric matrix $\mathbf{L}$ with $\mathbf{\Lambda}_{i,i} = \lambda_i, \lambda_1 \geq \cdots \geq \lambda_n$ and $\mathbf{U} = [\mathbf{u}_1, \ldots, \mathbf{u}_n]$. Let $\mathcal{U}_j \equiv span\{\mathbf{u}_1, \ldots, \mathbf{u}_j\}$. Assume $\kappa$-step Lanczos algorithm starts with vector $\nu$ and outputs the orthogonal $\mathbf{Q} \in \mathbb{R}^{n \times \kappa}$ and tridiagonal matrix $\mathbf{T} \in \mathbb{R}^{\kappa \times \kappa}$. For any $j$ with $1 < j < n$ and $\kappa > j$, we have $\|\mathbf{L} - \mathbf{Q}\mathbf{T}\mathbf{Q}^\top\|_F^2 \leq \sum_{j=i}^{1} \lambda_i^2 \left( \frac{sin(\nu, \mathcal{U}_i)\Pi_{k=1}^{j-1}(\lambda_k - \lambda_N)/(\lambda_k - \lambda_j)}{cos(\nu, \mathbf{u}_i)T_{K-i}(1 + 2\gamma_i)} \right)^2 + \sum_{N=i}^{j+1} \lambda_i^2$, where $T_{K-i}(x)$ is the Chebyshev Polynomial of degree $K - i$ and $\gamma_i = (\lambda_i - \lambda_{i+1})/(\lambda_{i+1} - \lambda_N)$.*

Based on Greenbaum's results Greenbaum (1989), we know that for our computed perturbed Lanczos algorithm exists an exact Lanczos algorithm but for a different matrix. Based on Theorem 4, we also cognize the upper bound of the low-rank approximator of the Lanczos algorithm. Therefore, the perturbed Lanczos algorithm, caused by the inaccuracy of the QR method for solving the least square equation, converges to the upper bound of the low-rank approximation of the symmetric matrix of the exact Lanczos algorithm.

| METRIC | CORA HR@100 | CITESEER HR@100 | PUBMED HR@100 | COLLAB HR@50 | VESSEL ROC-AUC |
|---|---|---|---|---|---|
| CN | 33.92 | 29.79 | 23.13 | 56.44 | 48.49 |
| AA | 39.85 | 35.19 | 27.38 | 64.35 | 48.49 |
| RA | 41.07 | 33.56 | 27.03 | 64.00 | N.A. |
| GCN | 66.79 | 67.08 | 53.02 | 44.75 | 43.53 |
| SAGE | 55.02 | 57.01 | 39.66 | 48.10 | 49.89 |
| NEO-GNN | 80.42 | 84.67 | 73.93 | 57.52 | N.A. |
| SEAL | 81.71 | 83.89 | 75.54 | 64.74 | 80.50 |
| NBFNET | 71.65 | 74.07 | 58.73 | OOM | N.A. |
| BUDDY | 88.00 | 92.93 | 74.10 | 65.94 | 55.14 |
| LLwLC | 91.44 | 93.40 | 83.10 | 66.86 | 79.60 |
| # PARAMS. | 0.019M | 0.018M | 0.024M | 0.026M | 0.019M |

Table 1: Results on link prediction benchmarks; LLwLC with only Neumann Constraints. The colors denote the best and second-best models. We train LLwLC with 10% of the VESSEL dataset.

## 4 EXPERIMENTS.

We extensively evaluate our proposed LLwLC against traditional heuristics (CN Barabási & Albert (1999), RA Zhou et al. (2009), AA Adamic & Adar (2003)); vanilla GNN (GCN Kipf & Welling (2016), SAGE Hamilton et al. (2017)); GNNs modifying the input graph of MPNN (SEAL Zhang & Chen (2018), NBFNet Zhu et al. (2021)); and GNNs with manual features as pairwise representations (Neo-GNN Yun et al. (2021), BUDDY Chamberlain et al. (2022)). Their results are from Chamberlain et al. (2022). We use four link prediction benchmarks. Three are the Planetoid citation networks Cora, Citeseer, and Pubmed (Yang et al. (2016)). The other one is ogbl-collab from the Open Graph Benchmark (Hu et al. (2020)). Dataset statistics and splits are shown in Appendix A.1. Baseline results for ogbl-collab are taken directly from the OGB leaderboard.

**Setup.** The learning rate in all experiments is 0.0001. All results are reported after 20 epochs. The MLP block applied on top of the Ritz values consists of two MLP layers (32 channels each) followed by the non-linearity (LeakyReLU) and the dropout. We fix the number of eigenpairs to be 10 at maximum, and in case less than the constant value, we pad it with 0. We implement all methods using PyTorch Paszke et al. and PyTorch Geometric Fey & Lenssen (2019). In the training phase, the training loss is the binary cross entropy (BCE) between the output prediction $\hat{y}$ and the ground-truth signal $y$; $\mathcal{L} : \text{BCE}(\hat{y}, y)$. Like SEAL Zhang & Chen (2018), we make the positive and negative testing data by randomly removing 10% of existing links from each dataset and randomly sampling the same number of nonexistent links, respectively. We also make the training data leveraging the remaining 90% of existing links and the same number of additionally sampled nonexistent links.

**Link Prediction Results.** Table 1 shows that our new eigenbasis LLwLC is a robust framework for link prediction and achieves a strong performance on link prediction benchmarks. With only 0.02M parameters, we outperform the previous models on the standard Planetoid dataset. This means the information provided by utilizing subgraph structures and encoding a subset of node relations leads to better results for predicting a link between two nodes in Cora, CiteSeer, and PubMed.

Also, we achieve state-of-the-art on the more complicated and real-world dataset Collab with only 0.02M parameters (compared to BUDDY Chamberlain et al. (2022) and SEAL Zhang & Chen (2018) with 1.10M and 0.50M parameters respectively). When we increase the number of blocks with only 0.03M parameters, we achieve 67.50 HR@50. In all experiments on this dataset, like the previous works, we only utilized 15% of the training dataset during the training phase to train our model. Also, on the Vessel dataset, we achieve on-par results with SEAL while only training with 10% of the training dataset and only 0.019M (less than 1% parameters).

**Time Complexity.** The time complexity of our method is $\mathcal{O}(\kappa E) + \mathcal{O}(k^2 n)$ for the outer loop (Lanczos algorithm) and the QR factorization, respectively, where $\kappa \ll n$ is the number of computed eigenvectors and $k \ll n$ is the number of linear constraints. Thus, we are linear w.r.t number of nodes. The comparison between the time complexity of LLwLC and the previous ones is represented in Table 3. The more interesting comparison would be with the higher-order universal approximator GNNs Maron et al. (2019a) with the time complexity of $\mathcal{O}(n^k)$. In Theorem 2, we show that if the reconstruction conjecture holds and we have all the $n - 1$ vertex-deleted subgraphs, then LLwLC is

| Metric
# Params. | Collab
HR@50
0.026M | Cora
HR@100
0.019M | PubMed
HR@100
0.021M |
|---|---|---|---|
| LanczosNet Liao et al. (2019) | 42.58 | 90.80 | 77.18 |
| LLwLC w. Neumann Constraints | 66.86 | 91.44 | 83.10 |
| LLwLC w. 10 Constraints | 69.40 | 93.10 | 82.28 |
| Two-block LLwLC w. Neumann Const. | 67.50 | # Params 0.035M | |

Table 2: Results on link prediction benchmark datasets with different numbers of vertex-deleted subgraphs as constraints. The color denotes the best model.

| Complexity | SEAL | BUDDY | NBFNet | LLwLC |
|---|---|---|---|---|
| Preprocessing | $\mathcal{O}(1)$ | $\mathcal{O}(lE(d+h))$ | $\mathcal{O}(1)$ | $\mathcal{O}(1)$ |
| Training (1 link) | $\mathcal{O}(Ed^2)$ | $\mathcal{O}(l^2h + ld^2)$ | $\mathcal{O}(Ed + nd^2)$ | $\mathcal{O}(\kappa E + k^2 n)$ |
| Inference | $\mathcal{O}(Ed^2)$ | $\mathcal{O}(l^2h + ld^2)$ | $\mathcal{O}(Ed + nd^2)$ | $\mathcal{O}(\kappa E + k^2 n)$ |

Table 3: Time Complexity. n and E are the number of nodes and edges with d-dimensional node features, l hops considered for propagation, and sketches with size h (in BUDDY). $\kappa$ is the number of eigenvectors and $k$ is the number of constraints.

| Metric | Cora
AUC | Citeseer
AUC | Pubmed
AUC |
|---|---|---|---|
| LanczosNet Liao et al. (2019) | 94.5% | 96.5% | 97.2% |
| LLwLC (w. **L** & **C**) | 97.0% | 98.1% | 98.3% |

Table 4: Planetoid Datasets results with **L** (with ground truth edge) with and w.o. Neumann constraints. The color denotes the best model.

universal approximator with worst case time complexity $\mathcal{O}(n^3)$ (as if $k = n - 1$). Also, based on the results of Barabási & Albert (1999), we know that in practice, only a few constraints are required for graph reconstruction. So, we are the first model to provide expressivity while having reasonable time complexity (in practice, linear to number of nodes).

**Ablation Studies.** To show the effect of applying the constraints, we make the comparison between the estimation of the basis with the ground truth Laplacian matrix **L** (edge given) and the approximation of the eigenbasis with the ground truth Laplacian matrix **L** and the Neumann constraints **C**. We show the results on three benchmark datasets in Table 4. We observe that injecting the Neumann constraints, **C**, significantly improves the results.

To analyse the effectiveness of our proposed layer, we study one block of LLwLC as shown in Figure 2 followed by a non-linearity, a global sort pooling layer, and a fully connected layer. The results are in Table 4. LLwLC (with Neumann constraints) beats the LanczosNet Liao et al. (2019) (without any linear constraints on the input graph) in all three benchmark datasets.

We conduct further experiments to see the effect of coupling more constraints to our architecture. The experiments in Table 2 show the effects of increasing the number of constraints by utilizing the vertex-deleted subgraphs. We observe that with only ten vertex-deleted subgraph constraints, we can achieve state-of-the-art and improve the results on benchmark datasets. Specifically, on the Collab dataset, the effect of constraints is remarkable (from HR@50 42.83 of LanczosNet without any linear constraints on the input graph to HR@50 69.40 with ten constraints of vertex deleted subgraphs. The results on the Cora dataset also show improvement from HR@100 90.80 to 93.10. The results are on par between PubMed with Neumann Constraints and Pubmed with ten vertex deleted subgraph constraints, while both improved the baseline results drastically). Our results align with the theory we provide in Theorem 2. Theorem 2 shows that having $n - 1$ deleted subgraphs as linear constraints leads to a universal approximator function. Favorably, we know that with three vertex-deleted subgraphs, almost all graphs are reconstructible Bollobás (1990), which our experiments support.

## 5 RELATED WORK.

**Link Prediction.** LP is a node-level task that by predicting missing or future links between pairs of nodes, becomes a crucial question on graph-structured data. The LP methods fall into three groups; heuristics Katz (1953); Newman (2001), unsupervised node embeddings or factorization methods Menon & Elkan (2011), and graph neural networks (GNNs) Gilmer et al. (2017). Heuristics and classical approaches are successful. Katz Katz (1953) measures the similarity between two nodes based on the weighted counts of paths between them. PageRank Page et al. (1998) computes the random walk probability. SimRank Jeh & Widom (2002) measures similarity considering the similarity between the neighbors of the two nodes. Despite their success, they are task-specific and

not generalizable. The embedding methods embed the nodes in the low dimension such that the distance between the nodes is kept. Despite their scalability, they are not inductive and cannot be tested in new graphs Perozzi et al. (2014); Grover & Leskovec (2016). Current LP approaches utilize GNNs, but their results are slightly worse than classical methods. The reason behind this is the limits on the expressivity power of the MPNNs, a commonly used approach in GNNs.

**MPNN Expressivity.** Measuring the expressivity power of GNNs involves addressing the graph isomorphism problem, which has no P solution (NP-intermediate). 1-WL test Weisfeiler & Leman (1968) solves graph isomorphism for most graph-structured data. MPNNs are at most as expressive as the 1-WL test Xu et al. (2019); Morris et al. (2019); Li & Leskovec (2022). The 1-WL test fails to distinguish only a few graph structures. This limitation is significant in real-world graph-structured applications. In particular, the 1-WL test fails to distinguish isomorphism between attributed regular graphs, measure the distance between different nodes, and count cycles Li & Leskovec (2022). The solutions fall into four categories: injecting random attributes , injecting deterministic distance attributes Zhang & Chen (2018), building higher order-GNNs Maron et al. (2019b), and subgraph-based approaches. Injecting random attributes makes the same features assigned to different nodes with the same substructure different, thus enabling the network to distinguish them. However, they are not deterministic; thus, the neural network has difficulty generalizing. Deterministic positional features (e.g. Zhang & Chen (2018)) argue that the incapability of the GNNs to encode the distance between nodes in the input graph raises the above issues and addresses them by injecting deterministic distance attributes. However, these methods assign different node features to isomorphic graphs and, thus, are not generalizable in inference time Li & Leskovec (2022). In parallel, a line of research discusses the expressivity of GNNs through subgraphs. For graph-level tasks, extracting subgraphs enhances expressivity, e.g., ESAN Bevilacqua et al. (2022) proposed selecting a bag of subgraphs. For the LP task, Surel Yin et al. (2022) and Surel+ Yin et al. (2023) also encode more expressive features by considering subgraphs. However, they have to do it system level and offline. Leveraging this, several graph convolution-based deep network models have been proposed Kipf & Welling (2016); Susnjara et al. (2015); Liao et al. (2019).

**Subgraph GNNs for Link Prediction.** The first subgraph-based link prediction architecture is the Weisfeiler Lehman Neural Machine (WLNM) Zhang & Chen (2017). SEAL Zhang & Chen (2018) improves WLNM by replacing fully connected with graph convolutional layers and encoding positional features by proposing the DRNL approach to improve the node labeling instead of utilizing the Weisfeiler Lehman coloring. SEAL Zhang & Chen (2018) proves the sufficiency of information in two hops away subgraphs by defining $\gamma$-decaying heuristics, which unifies the classical methods to $\eta \sum_{l=1}^{\infty} \gamma^l f(x, y, l)$. $0 \leq \gamma \leq 1$ is a decaying factor, $\eta \geq 0$ is a constant or a function of $\gamma$ that is upper bounded by a constant, and $f$ is a non-negative function of $x, y, l$ under the given network. Based on the $\gamma$-decaying heuristic, it shows that the crucial information for the classical methods (e.g., Katz, PageRank, and SimRank) lies in two hops away nodes (given $0 \leq \gamma \leq 1$ the increase to $l$ in $\gamma$-decaying heuristic makes the higher orders less effective). SEAL Zhang & Chen (2018) does not encode the pairwise node representation. To compute the features without computational overhead, Neo-GNN ( Yun et al. (2021)) and BUDDY ( Chamberlain et al. (2022)) decouple the pairwise representation from node representation learning. By leveraging the extracted manual features as pairwise representations, they only run MPNN on the original graph. While this leads to better scalability, these pairwise representations are oversimplified.

## 6 CONCLUSION.

We propose a novel eigenbasis to encode the linear constraints (relations between nodes/edges in the input graph) explicitly and efficiently. Thus, this new basis can address the limitations of the MPNNs, e.g., their inability to distinguish between some graphs, e.g., between the k-regular graphs. We specifically propose the Neumann constraints, which encode the edge relation between one-hop and two-hop away subgraphs (this specific constraint can distinguish between k-regular graphs). Also, it encodes the relation between the one-hop-away nodes. We also investigated the effect of encoding the vertex-deleted subgraph constraints, which leads to improvement in benchmark datasets. We provide a rigorous proof of convergence of the LLwLC eigenbasis. Also, we show that LLwLC is a universal approximator while its complexity is linear w.r.t number of nodes. In future work, we will investigate the effect of learning the linear constraints between nodes/edges of the input graph.

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

Table 5: Properties of link prediction benchmarks. Confidence intervals are +/- one standard deviation. Splits for the Planetoid datasets are random and Collab uses the fixed OGB splits.

|  | CORA | CITESEER | PUBMED | COLLAB |
|---|---|---|---|---|
| # NODES | 2708 | 3327 | 18717 | 235868 |
| # EDGES | 5278 | 4676 | 44,327 | 1,285,465 |
| SPLITS | RAND | RAND | RAND | TIME |
| AVG DEG | 3.9 | 2.74 | 4.5 | 5.45 |
| AVG DEG | 15.21 | 7.51 | 20.25 | 29.70 |
| 1-HOP SIZE | 12+/-15 | 8+/-8 | 12+/-17 | 99 +/-251 |
| 2-HOP SIZE | 127+/-131 | 58+/-92 | 260+/-432 | 115+/-571 |

# A    APPENDIX

## A.1    DATASET DETAILS.

The properties of the link prediction benchmarks used throughout our evaluations are presented in Table 5.

## A.2    THEORETICAL ANALYSES

**Proof of Theorem 3.** It is the immediate result of the following theorem discussed in Golub et al. (2000).

**Theorem 5.** *Let $\mathcal{U}$ and $\tilde{\mathcal{U}}$ be the eigenspaces corresponding to the smallest eigenvalues $\lambda$ and $\tilde{\lambda}$ of the symmetric matrices $\mathbf{A}$ and $\tilde{\mathbf{A}} = \mathbf{A} + \mathbf{E}$, respectively. Then*

*1. For any $\mathbf{u} \in \mathcal{U}$ and $\tilde{\mathbf{u}} \in \tilde{\mathcal{U}}$ with $\|\mathbf{u}\|_2 = \|\tilde{\mathbf{u}}\|_2 = 1$,*

$$\tilde{\mathbf{u}}^\top \mathbf{E} \tilde{\mathbf{u}} \leq \tilde{\lambda} - \lambda \leq \mathbf{u}^\top \mathbf{E} \mathbf{u}.$$

*2. For any $\tilde{\mathbf{u}} \in \tilde{\mathcal{U}}$ with $\|\mathbf{u}\|_2 = 1$, there exists $\mathbf{u} \in \mathcal{U}$ with $\|\tilde{\mathbf{u}}\|_2 = 1$ such that*

$$\beta \leq \|\mathbf{u} - \tilde{\mathbf{u}}\|_2 \leq \beta(1 + \frac{\beta^2}{1 + \sqrt{2}}),$$

*where $\beta$ satisfies*

$max\{0, \frac{\|\mathbf{E}\tilde{\mathbf{u}}\|_2 - |\tilde{\lambda} - \lambda|}{\tilde{d}_{max}}, \frac{\|\mathbf{E}\mathbf{u}\|_2 - |\tilde{\lambda} - \lambda|}{d_{max}}\} \leq \beta \leq min\{\frac{\|\mathbf{E}\hat{\mathbf{u}}\|_2}{\tilde{d}_{min}}, \frac{\|\mathbf{E}\mathbf{u}\|_2}{d_{min}}\}$

*with*

$\tilde{d}_{min} = min\{|\tilde{\lambda} - \lambda(\mathbf{A})| | \lambda(\mathbf{A}) \neq \lambda\},$

$\tilde{d}_{max} = max\{|\tilde{\lambda} - \lambda(\mathbf{A})| | \lambda(\mathbf{A}) \neq \lambda\},$

$d_{min} = min\{|\lambda - \lambda(\tilde{\mathbf{A}})| | \lambda(\tilde{\mathbf{A}}) \neq \tilde{\lambda}\},$

$d_{max} = max\{|\lambda - \lambda(\tilde{\mathbf{A}})| | \lambda(\tilde{\mathbf{A}}) \neq \tilde{\lambda}\}.$

**Proof of Theorem 4** As addressed by Liao et al. (2019); Parlett (1980), we have $\mathbf{LQ} = \mathbf{QT}$ from the Lanczos algorithm. Therefore,

$$\|\mathbf{L} - \mathbf{QTQ}^\top\|_F^2 = \|\mathbf{L} - \mathbf{LQQ}^\top\|_F^2 = \|\mathbf{L}(\mathbf{I} - \mathbf{QQ}^\top)\|_F^2$$

Let $\mathbf{P}_{\mathbf{Q}}^\perp \equiv \mathbf{I} - \mathbf{QQ}^\top$, the orthogonal projection onto the orthogonal complement of subspace span$\{\mathbf{Q}\}$. Relying on the eigendecomposition we have,

$$\|\mathbf{L} - \mathbf{Q}\mathbf{T}\mathbf{Q}^\top\|_F^2 =$$
$$\|\mathbf{U}\mathbf{\Lambda}\mathbf{U}^\top(\mathbf{I} - \mathbf{Q}\mathbf{Q}^\top)\|_F^2 =$$
$$\|\mathbf{\Lambda}\mathbf{U}^\top(\mathbf{I} - \mathbf{Q}\mathbf{Q}^\top)\|_F^2 =$$
$$\|(\mathbf{I} - \mathbf{Q}\mathbf{Q}^\top)\mathbf{U}\mathbf{\Lambda}\|_F^2 =$$
$$\|[\lambda_1\mathbf{P}_\mathbf{Q}^\perp\mathbf{u}_1, \dots, \lambda_N\mathbf{P}_\mathbf{Q}^\perp\mathbf{u}_N]\|_F^2,$$

where we use the fact that $\|R\mathbf{A}\|_F^2 = \|\mathbf{A}\|_F^2$ for any orthogonal matrix $\mathbf{R}$ and $\|\mathbf{A}\|_F^2 = \|\mathbf{A}\|_F^2$. Note that for any $j$ we have,

$$\|[\lambda_1\mathbf{P}_\mathbf{Q}^\perp\mathbf{u}_1, \dots, \lambda_N\mathbf{P}_\mathbf{Q}^\perp\mathbf{u}_N]\|_F^2 = \sum_{i=1}^N \lambda_i^2\|\mathbf{P}_\mathbf{Q}^\perp\mathbf{u}_i\|^2 \le \lambda_i^2\|\mathbf{P}_\mathbf{Q}^\perp\mathbf{u}_i\|^2 + \sum_{i=j+1}^N \lambda_i^2,$$

where we use the fact that for any $i$, $\|\mathbf{P}_\mathbf{Q}^\perp\mathbf{u}_i\|^2 = \|\mathbf{u}_i\|^2 - \|\mathbf{u}_i - \mathbf{P}_\mathbf{Q}^\perp\mathbf{u}_i\|^2 \le \|\mathbf{u}_i\|^2 = 1$. Note that we have span$\{\mathbf{Q}\}$ = span$\{\nu, \mathbf{L}\nu, \dots, \mathbf{L}_{K-1}\nu\} \equiv \kappa_K$ from the Lanczos algorithm. Therefore, we have,

$$\|\mathbf{P}_\mathbf{Q}^\perp\mathbf{u}_i\| = |\sin(\mathbf{u}_i, \kappa_K)| \le |\tan(\mathbf{u}_i, \kappa_K)|.$$

We finish the proof by applying the above lemma with $\mathbf{A} = \mathbf{L}$.

**Proof of Theorem 1.**

**Definition 1** (Graph isomorphism and automorphism). *Let $G_1 = (V_1, E_1)$, $G_2 = (V_2, E_2)$ be two simple graphs. An isomorphism between $G_1, G_2$ is a bijective map $\Phi : V_1 \to V_2$ which preserves adjacencies, that is: $\forall u, v \in V_1 : (u, v) \in E_1 \iff (\Phi(u), \Phi(v)) \in E_2$. If $G_1 = G_2$, $\Phi$ is called an automorphism Chamberlain et al. (2022).*

BUDDY Chamberlain et al. (2022) can distinguish between edges in different orbits (solves the node automorphism problem) (see Figure 3). We also can address the problem because the one hop-away nodes, two hop-away nodes, and the Neumann eigenvalue constraints are different in different orbits of the $C_6$ graph.

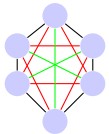

Figure 3: The $C_6$ graph shows three different orbits with three different colors. Both BUDDY Chamberlain et al. (2022) features and our proposed features can distinguish them.

BUDDY Chamberlain et al. (2022) features cannot improve the MPNN features to discriminate between the 2-regular graphs Figure 4. Because the 1-hop and 2-hop structure features for nodes 1 and 2 are the same in both graphs. On the contrary, our proposed linear constraints (induced subgraphs represented in constraint-1 in Figure 4) for the query nodes (1 and 2) in $G_1$ differ from those in $G_2$. Thus, our proposed method is more expressive than all the previously proposed methods.

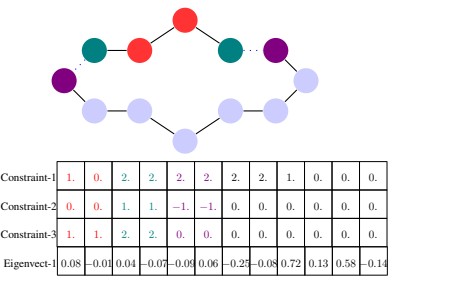 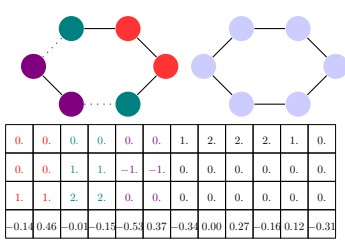

Figure 4: Two k-regular subgraphs with their corresponding Neumann constraints and Neumann eigenvectors and linear constraints (induced subgraphs). For the two k-regular subgraphs, the MPNN makes the same tree, while our proposed eigenbasis (*LLwLC*) can distinguish between them.

