# OpenReview forum: "LINK PREDICTION USING NEUMANN EIGENVALUES"
_ICLR.cc/2024/Conference — Submitted to ICLR 2024_

### Official Review · Reviewer_ZLDq · 2023-10-20

**Soundness:** 3 good
**Presentation:** 2 fair
**Contribution:** 3 good
**Rating:** 8
**Confidence:** 3

**Summary:**

This paper focuses on link prediction tasks in GNNs. It addresses the issue of traditional GNNs being unable to distinguish between some graphs due to their message-passing structures. The authors tackle this challenge by proposing a novel learnable Lanczos algorithm with linear constraints, LLwLC. Specifically, the authors extend the Lanczos algorithm and project the approximated eigenvectors with a Neumann constraint matrix to efficiently encodes the relation between nodes. The experimental results demonstrate the effcitiveness of the proposed algorithem in link prediction tasks.

**Strengths:**

+ The paper is technically solid and introduces a novel spectral GNN structure, offering an insightful exploration of utlizing linear constraints on subgraph structures in eigenspaces.
+ The proposed LLwLC has the capability to distinguish the k-regular graphs, which is an improvement over the MPNNs.
+ Experimental results show the effectiveness of the proposed LLwLC in link prediction tasks.

**Weaknesses:**

- The language and presentation of the paper could be further improved, particularly in figures and visual representations. (e.g. Figure 2).
- While the paper is designed for link prediction tasks, the motivation behind applying LLwLC in link prediction is not well-explained, especially given that the propagation process appears to be a learnable spectral graph filter. It would be helpful to clarify why LLwLC is particularly suited for link prediction tasks.
- Though the authors have demonstrated that the expressiveness LLwLC propagation is superior to MPNNs on k-regular graphs, they are still a limited subset of all the 1-WL isomorphism graphs.
- The implementation of the proposed model is not accessible, which makes it difficult for others to reproduce the experiments and further validate the results.

**Questions:**

1. What motivated the application of LLwLC in link prediction tasks? How does the model's performance compare in other node-wise downstream tasks, such as node classification?
2. Can you proof the superiority of LLwLC over MPNNs on a wider range of 1-WL isomorphism graphs?
3. Can you provide information about the computational complexity of the proposed LLwLC algorithm?

---

> ### Author Response · Authors · 2023-11-14
>
> # Reviewer ZLDq
> We thank the reviewer for the valuable feedback and for recognizing the motivation and the effectiveness of our work. In the following, we will clarify the raised questions.
> ### Q1: Why Link Prediction task
> Theoretically, we propose a new spectral GNN that can be applied to many problem settings. Essentially, it is an eigenbasis capable of encoding the linear constraints, which improves the expressivity.  We focus on the link prediction task, which mainly its success depends on counting the sub-structures and triangles, which can be resolved by discriminating between the k-regular graphs, (what LLwLC is capable of doing it). We expect that LLwLC performs well in other tasks, but we did not conduct such experiments.
> ### Q2: Expressivity
>
> The sufficient conditions under which LLwLC can solve graph isomorphism entails that LLwLC is a universal approximator of functions defined on graphs [1]. Given that we can encode any subgraph into our eigenbasis, we can examine whether there exists a specific substructure collection that can completely characterise each graph. By the reconstruction conjecture we know that if we have all the $n − 1$ vertex-deleted subgraphs, then we can reconstruct the graph. Consequently, if the reconstruction conjecture holds and the substructure collection contains all graphs of size $k = n-1$ then LLwLC can distinguish all non-isomorphic graphs of size $n$ and is therefore universal. Based on [2], almost every graph has reconstruction number three. This is consistent with our experimental results, where we observed that certain small substructures such as Neumann constraints, significantly improve the results.
>
>
> [1] Chen, Zhengdao and Villar, Soledad and Chen, Lei and Bruna, Joan, "On the Equivalence between Graph Isomorphism Testing and Function Approximation with GNNs.", NeurIPS, 2019.
> [2] Bollobas, Bela, "Almost Every Graph Has Reconstruction Number Three.", Journal of Graph Theory, 1990.
>
>
> ### Q3: Computational Complexity
>
> The time complexity of extracting the subgraphs is at most O(n) (though as
> we discussed we can do it with only three subgraphs). The time complexity of our method, given the sparsity of graph Laplacian matrix, is $\mathcal{O}(\kappa E)+ \mathcal{O}(k^2n)$ for the outer loop (Lanczos algorithm) and the QR factorization, respectively, where $\kappa \ll n$ is the number of computed eigenvectors and $k \ll n$ is the number of linear constraints. Thus, we are linear w.r.t number of nodes.

---

> > ### Comment · Reviewer_ZLDq · 2023-11-21
> >
> > I appreciate the authors' efforts in addressing my concerns and providing relative references and I will adjust my evaluation accordingly. However, I wish to restate that there is still room for improvement in the presentation of this paper (e.g. Figure2, Table1~3) and it will be a much more readable paper if authors would make an effort on that. Despite this, I believe it's still an insightful idea to work on spectral GNNs from the view of  Lanzcos algorithm and I'll support this paper.

---

> > > ### Author Response · Authors · 2023-11-22
> > >
> > > We highly appreciate your time and effort in reviewing our paper and providing excellent questions and feedback to improve our work further. We updated the paper to improve on the presentation issues.

---

### Official Review · Reviewer_eeNp · 2023-10-31

**Soundness:** 2 fair
**Presentation:** 2 fair
**Contribution:** 2 fair
**Rating:** 6
**Confidence:** 3

**Summary:**

This paper presents a new algorithm to perform Link Prediction/Classification. The main idea is to encode subgraph information via a captured eigenbasis of a constrained Laplacian matrix. According to the paper, the new basis makes the features more expressive by explicitly encoding the linear constraints over the graph. Numerical experiments verify the effectiveness of the new algorithm.

**Strengths:**

-) The paper considers important and relevant problems.
-) Numerical experiments suggest that the proposed algorithm can attain high accuracy versus other competitive approaches.

**Weaknesses:**

-) The paper is not very clear. I had to repeat many sentences several times in order to understand what the authors aimed to state.
The language also needs improvements.
-) Wall-clock time comparisons are absent.
-) The new algorithm seems to be a relative minor extension, especially the parts related to Lanczos and its theoretical analysis. What is the novel theoretical analysis? Most results are standard from what I know in the subject. Please elaborate more on these two fronts.

**Questions:**

-) "Proposition. If we start simple Lanczos with ν ∈ N (C⊤), then qj ∈ N (C⊤) for all j." This is true only in theory as in practice round-off errors re-introduce the deflated direction. This is well-known in numerical linear algebra and Lanczos is very rarely used without some form of restart precisely for this reason.

-) In Figure 2, purple box: you mention "svd(T)" but it is clear you mean "eig(T)".

-) The statement about the orthogonal projector is wrong. C(C^TC)^{-1} holds for the case of independent rows in C; instead, in your case you have linearly independent columns thus it should be  (C^TC)^{-1}C^T. You can verify then that P=I-C(C^TC)^{-1}C^T is what you want, and not P=I-CC(C^TC)^{-1} which the math indicate in your paper (the multiplication CC is not even defined).

-) The discussion in Sections 4.2 and 4.3 are basically straightforward and read as textbook-style material. I would remove both.

-) The results indicate superior performance for LLwLC but no timings are shown. In other words, higher accuracy should not be accompanied by high costs. Indeed, the authors do a lot more work per Lanczos step via LSQR -- this is essentially applying Lanczos
with shift-and-invert.

-) The keyword 'Neumann' keeps appearing only to be explained on page 7.

---

> ### Author Response · Authors · 2023-11-14
>
> # Reviewer eeNp
> We thank the reviewer for the insightful comments and will clarify the raised questions in the following.
>
> ### W1: Theoretical Analysis of LLwLC.
> We provide a new spectral GNN which encodes the subgraph structures into the eigenbasis. We encode relations between a "set of nodes/edges": we write each subgraph as a linear constraint, formulate it as an eigenvalue problem with linear constraints, and compute the eigenvectors accordingly. Encoding link representation (the same as the first constraint in Neumann constraints in Eq. 3 of the paper) in graph-structured data gives the power to count triangles and substructures. The sufficient conditions under which LLwLC can solve graph isomorphism entails that LLwLC is a universal approximator of functions defined on graphs [1]. Given that we can encode any subgraph into our eigenbasis, we can examine whether there exists a specific substructure collection that can completely characterise each graph. By the reconstruction conjecture[2], we know that if we have all the $n − 1$ vertex deleted subgraphs, then we can reconstruct the graph. Consequently, if the reconstruction conjecture holds and the substructure collection contains all graphs of size $k = n-1$, then LLwLC can distinguish all non-isomorphic graphs of size $n$ and is therefore universal. Based on [3], almost every graph has reconstruction number three. This is consistent with our experimental results, where we observed that certain small substructures such as Neumann constraints (or in the other ablation study with only ten vertex-deleted subgraphs) significantly improve the results. While, in theory, LLwLC can be applied to many problem settings, we focus on the challenging link prediction task.
>
> [1] Chen, Zhengdao and Villar, Soledad and Chen, Lei and Bruna, Joan, "On the Equivalence between Graph Isomorphism Testing and Function Approximation with GNNs.", NeurIPS, 2019.
> [2] Ulam, Stanislaw M, "A Collection of Mathematical Problems", 1960.
> [3] Bollobas, Bela, "Almost Every Graph Has Reconstruction Number Three.", Journal of Graph Theory, 1990.
>
>
> ### Q1 & Q4:
> Thank you for the comment. As you mentioned in Q1, the round-off errors might raise numerical issues, which must be addressed. Actually, the numerical round-off errors are studied in sections 4.2 and 4.3and  we prove that our proposed LLwLC has a convergence (numerical round-off errors are analysed here).
>
> In section 4.2 (section 3.3 in the revision), we theoretically analyze the effect of numerical roundoff errors, which lead to a perturbed tridiagonal matrix. Theorem 1 shows the error bounds of the computed perturbed tridiagonal matrix.
> In section 4.3 (section 3.3 in the revision), we utilize Greenbaum’s results, which mention that for our computed perturbed Lanczos algorithm exists, an exact Lanczos algorithm but for a different matrix.
> Also, Theorem 2 cognize the upper bound of the low-rank approximator of the Lanczos algorithm.
>
> So, based on these three facts, we conclude that LLwLC has convergence, and the effect of practical round-off errors in the LLwLC is accurately investigated.
>
> ### Q2: svd(T) in Fig. 2
> In this step, we decompose the tridiagonal matrix $\mathbf{T}$ using eigendecomposition.
> ### Q3: Orthogonal Projector
> Thank you for the detailed correction.
>
> ### Q5: Timings
> The average time complexity is O(n). The time complexity of our method, given the sparsity of graph Laplacian matrix, is $\mathcal{O}(\kappa E)+ \mathcal{O}(k^2n)$ for the outer loop (Lanczos algorithm) and the QR factorization, respectively, where $\kappa \ll n$ is the number of computed eigenvectors and $k \ll n$ is the number of linear constraints. Thus, we are linear w.r.t number of nodes.
> ### Q6: Neumann explanation
> We revised the introduction to clarify that the only necessary constraint for LLwLC is that matrix C be full column rank. So, we can encode any linear constraint in our matrix. One possibility which is quite useful is the Neumann constraints, in which the first constraint is the link representation between the links in one-hop and two-hop away nodes (link representation), and the second constraint is the one-hop away subgraph.

---

> > ### Comment · Reviewer_eeNp · 2023-11-22
> >
> > Given the minor improvements added by the authors, and to be fair to their effort, I will change my score from 5 to 6; however I do still think that this paper still needs improvement on various fronts (editing as well).  For example, the caption in Table 1 appears above the table but in Tables 2 3 and 4 appears below. In addition, timing/computational asymptotic complexity is an OK effort, but I really did look for actual wall-clock time comparisons (also, linear complexity with respect to the # of nodes is expected anyway).

---

> ### Author Response · Authors · 2023-11-23
>
> We highly appreciate your time in reviewing our paper and providing constructive feedback to improve our work further. We updated the paper to improve on the presentation issues.

---

### Official Review · Reviewer_NmSE · 2023-11-05

**Soundness:** 2 fair
**Presentation:** 1 poor
**Contribution:** 2 fair
**Rating:** 3
**Confidence:** 4

**Summary:**

In link prediction, because existing works of encoding link representation are prohibitively expensive，this paper proposes a novel light learnable eigen basis to encode the link representation and induced subgraphs efficiently and explicitly. Experiments shows the efficacy of the proposed method, and achieve the SOTA in benchmark datasets.

**Strengths:**

The proposed method is effective on some common datasets.

**Weaknesses:**

1. The writing of this paper is poor. For example, I cannot understand the motivation of this article from the introduction section. On the contrary, there is too much content about existing work in the introduction. Still, I cannot see the connection between these existing methods and the proposed method from this section.
2. What is the relationship between LLwLC and link prediction? LLwLC is only designed as a matrix factorization method for GNN.
3. This paper states that LLwLC has improved the efficiency of encoding link representations, but no specific form of link representation is mentioned in Section 4.
4. The conclusion of this article is not clear and requires a comparison between different methods, such as NBFNet and Seal, in terms of their link representation approach and complexity.
5. It is necessary to compare the running time to show the efficiency of the proposed method.
6. It is better to compare more methods on Ogbl-Collab. Currently, only LanczosNet is compared.

**Questions:**

See weakness

---

> ### Author Response · Authors · 2023-11-14
>
> # Reviewer NmSE
> We thank the reviewer for the feedback and address the raised questions in the following.
> ### Q1: Clarity
> In the introduction, we summarized the MPNN expressivity limitations, mainly their inability to count cycles and the three main directions in the literature to address this problem. Then, we propose our novel spectral GCN, which explicitly utilizes the subgraph structures in eigenspaces to mitigate the existing limitations of GNN expressivity specifically for the task of link prediction where we can accurately show the effectiveness of our LLwLC (as discussed in Q2). We could shorten the discussion of the limitations if the reviewer thinks it would improve the clarity.
>
> ### Q2: Relation between LLwLC and Link Prediction
> We provide a new spectral GNN which encodes the subgraph structures into the eigenbasis. We encode relations between a "set of nodes/edges": we write each subgraph as a linear constraint, formulate it as an eigenvalue problem with linear constraints and compute the eigenvectors accordingly. While, in theory, LLwLC can be applied to many problem settings, we focus on the challenging link prediction task, which mainly its success depends on counting the sub-structures and triangles, which can be resolved by discriminating between the k-regular graphs (what LLwLC is capable of doing it). (Also, the expressivity power of LLwLC is discussed in Q4).
> ### Q3: Link representation
> The first constraint of the Neumann constraint is the link representation constraint.
> ### Q4: Differences to NBFNet & SEAL
> The main difference between our work and SEAL & NBFNet is that we consider the relation between "a set of nodes (an induced subgraph encoded as a linear constraint) or link representations between a group of nodes", which leads to more expressivity compared to MPNNs (Specifically, encoding link representation (the same as the first constraint in Neumann constraints in Eq. 3 of the paper) in graph-structured data gives the power to count triangles and substructures). The time complexity of LLwLC, given the sparsity of the graph Laplacian matrix, is in average linear w.r.t number of nodes. The more important comparison is concerning the expressivity of LLwLC.  The sufficient conditions under which LLwLC can solve graph isomorphism entails that LLwLC is a universal approximator of functions defined on graphs [1]. Given that we can encode any subgraph into our eigenbasis, we can examine whether there exists a specific substructure collection that can completely characterise each graph. By the reconstruction conjecture[2], we know that we can reconstruct the graph if we have all the $n − 1$ vertex deleted subgraphs. Consequently, if the reconstruction conjecture holds and the substructure collection contains all graphs of size $k = n-1$, then LLwLC can distinguish all non-isomorphic graphs of size $n$ and is therefore universal. Based on [3], almost every graph has reconstruction number three. This is consistent with our experimental results, where we observed that certain small substructures such as Neumann constraints (or in the other ablation study with only 10 vertex-deleted subgraphs) significantly improve the results.
>
> [1] Chen, Zhengdao and Villar, Soledad and Chen, Lei and Bruna, Joan, "On the Equivalence between Graph Isomorphism Testing and Function Approximation with GNNs.", NeurIPS, 2019.
> [2] Ulam, Stanislaw M, "A Collection of Mathematical Problems", 1960.
> [3] Bollobas, Bela, "Almost Every Graph Has Reconstruction Number Three.", Journal of Graph Theory, 1990.
>
> ### Q5: Time Complexity
> The time complexity of our method, given the sparsity of graph Laplacian matrix, is $\mathcal{O}(\kappa E)+ \mathcal{O}(k^2n)$ for the outer loop (Lanczos algorithm) and the QR factorization, respectively, where $\kappa \ll n$ is the number of computed eigenvectors and $k \ll n$ is the number of linear constraints. Thus, we are linear w.r.t number of nodes.
>
>
> ### Q6: Comparison OGBL-Collab
> Table 1 of the original paper actually contains the comparison to other methods. We just separated LanczosNet and LLwLC to show the effect of our proposed method in direct comparison to LanczosNet.

---

### Official Review · Reviewer_8BZ5 · 2023-11-06

**Soundness:** 2 fair
**Presentation:** 1 poor
**Contribution:** 2 fair
**Rating:** 3
**Confidence:** 5

**Summary:**

This paper generalizes LanczosNet for representation learning of induced subgraphs for link prediction by formulating subgraphs as Neumann boundary conditions of the eigenvalue problem, which is solved by the Lanczo algorithm with linear constraints. As a result, a new expressive feature based on Neumann constraints is proposed to mitigate the limited expressive power of MPNNs. The proposed model LLwLC is evaluated on four citation networks and shows its effectiveness.

**Strengths:**

- A new type of structural feature is proposed, which shows its effectiveness in enhancing the MPNN framework.
- The proposed model builds a connection between spectral graph theory, numerical analysis, and subgraph-GNN.

**Weaknesses:**

- The benefit of introducing the proposed Neumann feature is debatable for link prediction. It shows that the node automorphism issue is addressed, but the model still depends on instance features for more expressiveness. Particularly, SubGNNs with simple features of low computation cost (e.g. zero-one labeling or DRNL) are already good enough, and the ablation study does not exactly separate the contribution between them.
- LLwLC is claimed to be lightweight and efficient, However, neither the theoretical complexity nor the wall-clock time is provided. Meanwhile, the proposed model is subgraph-based, which still suffers from the computation overhead of subgraph extraction and raises my concerns over its scalability on larger graphs (other OGB LP benchmark datasets).
- The organization of the paper needs some polishing, especially clearly establishing the connection between Neumann constraints and subgraphs, better illustration of figures, and differentiating the content/contribution from [1] for the result of numerical analysis. More comprehensive experiments including large-scale datasets, stronger baselines, clearer ablation studies, and runtime comparison are needed.

[1] Golub, Gene H., Zhenyue Zhang, and Hongyuan Zha. "Large sparse symmetric eigenvalue problems with homogeneous linear constraints: the Lanczos process with inner–outer iterations." Linear Algebra and its Applications 309.1-3 (2000): 289-306.

**Questions:**

* Is the proposed Neumann features only beneficial for the node automorphism issue? Can it also address the limitation of MPNN in counting sub-structures and triangles (other than the expressiveness inherited from subgraph-based models)?
* SEAL and BUDDY are not necessarily limited to 2-hop neighborhoods (the former is limited by scalability, the latter can be applied to arbitrary $k$ order). Can the Neumann basis be applied to more than 2-hop?
* It would be great to have a detailed example of Fig. 1 to show the construction of linear constraints.
* What is the complexity of the proposed method to extract a bag of subgraphs, obtain $\delta S$, and construct $C$ for each link?
* Technical detail:
  - Sec. 2 Related Work, the embedding methods are (not) inductive?
  - The iteration of sample Lanczos from [1] starts from $k=0$ while $j=1$ in LanczosNet. The inconsistency causes confusion about the iteration of the Lanczos Algorithm in introduced Sec 3, especially the coefficients of $q_{j-1}$, $q_{j+1}$.
  - What does it mean that “SEAL does not encode the pairwise node representation”? SEAL converts the link prediction as subgraph classification, which encodes the induced subgraph of two queried nodes with their distance labels. A similar framework is also adopted by LLwLC.

---

> ### Author Response · Authors · 2023-11-14
>
> # Reviewer 8BZ5
> We thank the reviewer for the insightful comments. There were a few questions which we want to clarify in the following.
>
> ### Q1: Benefits of Linear Constraints (Neumann constraints)
> We encode relations between a "set of nodes/edges": we write each subgraph as a linear constraint and formulate it as an eigenvalue problem with linear constraints, and compute the eigenvectors accordingly. Encoding link representation (the same as the first constraint in Neumann constraints in Eq. 3 of the paper) in graph-structured data gives the power to count triangles and substructures.  The sufficient conditions under which LLwLC can solve graph isomorphism entails that LLwLC is a universal approximator of functions defined on graphs [1]. Given that we can encode any subgraph into our eigenbasis, we can examine whether a specific substructure collection can completely characterise each graph. By the reconstruction conjecture[2], we know that if we have all the $n − 1$ vertex deleted subgraphs, we can reconstruct the graph. Consequently, if the reconstruction conjecture holds and the substructure collection contains all graphs of size $k = n-1$, then LLwLC can distinguish all non-isomorphic graphs of size $n$ and is therefore universal. Based on [3], almost every graph has reconstruction number three. This is consistent with our experimental results, where we observed that certain small substructures such as Neumann constraints (or in the other ablation study with only ten vertex-deleted subgraphs) significantly improve the results.
>
> [1] Chen, Zhengdao and Villar, Soledad and Chen, Lei and Bruna, Joan, "On the Equivalence between Graph Isomorphism Testing and Function Approximation with GNNs." NeurIPS, 2019.
> [2] Ulam, Stanislaw M, "A Collection of Mathematical Problems", 1960.
> [3] Bollobas, Bela, "Almost Every Graph Has Reconstruction Number Three.", Journal of Graph Theory, 1990.
>
> ### Q2: More than 2-hop neighborhoods
> LLwLC is not limited to 2-hop neighborhoods (with an average time complexity of O(n)). In theory, the only requirement that matrix C must satisfy is to be full column rank.
>
> ### Q3: Detailed Explanation of Fig. 1
> The first Neumann constraint is the relation between nodes in 1-hop-away nodes and two-hop-away nodes $\Sigma_{y \in S, y\sim x} (f(x) - f(y)) = 0$ (edge representation between them). We represent it in the first row. So, if we multiply the first constraint with the eigenvector ($C^Tf = 0; (Left: -0.40 + 0.61 + 0.03 - 0.25 \approx 0$,  Right: $-0.39 - 0.20 + 2 \times (0.29) \approx 0$)).
>
>
> The second Neumann constraint is 1-hop-away nodes of the subgraph which is encoded by the degree of nodes in the subgraph $\Sigma f(x)d_x = 0$. We show it in the second row. So, if we multiply the second row with the eigenvector in the third row ($C^Tf=0$; Left: $2 \times 0.24 - 2\times 0.46 - 2\times 0.40 + 2 \times 0.61 = 0$ Right: $-2 \times 0.39 + 2 \times 0.23 + 2\times 0.36 - 2 \times 0.20 = 0$).
>
> When we compute the eigenvectors by projecting to the null space of the constraints, we force the eigenvectors to be aware of the relation between a set of nodes/edges explicitly. So, the third row is one of the computed eigenvectors for the graphs which satisfy the constraints explicitly.
>
>
> ### Q4: Time Complexity of "extracting bag of subgraphs"
>
> Theoretically, regarding the time complexity, as we discussed in question 1,  based on the reconstruction conjecture [1] we know that if we have all the $n-1$ vertex deleted subgraphs, then we can reconstruct the graph. So, worst-case time complexity is $\mathcal{O}(n)$, but as we discussed, theoretically[2] and experimentally, we can do it with a much less number of constraints.
>
>
> [1]  Ulam, Stanislaw M, "A Collection of Mathematical Problems", 1960.
> [2] Bollobas, Bela, "Almost Every Graph Has Reconstruction Number Three." Journal of Graph Theory, 1990.
>
> ### Q5: Technical Detail
>
> Thanks for the detailed & valuable technical comments.
> * Q5.1: Thanks for the correction. Of course, they cannot be applied to the inductive setting.
> * Q5.2: We will change the iteration indexing such that both start at 0.
> * Q5.3 & W1: As mentioned in the answer to Q1, SEAL does not utilize the relation between a set of nodes/edges, whereas the goal of LLwLC is to encode the relation between a set of nodes into the eigenbasis  (SEAL ignores the pairwise relation between nodes). Also, regarding the effect of DRNL, as discussed in W1, please notice that the experiments with the LanczosNet in Tables 2 and 3 (specifically, Collab results) also utilize the DRNL features. However, their results are drastically worse than our novel LLwLC.
>
> ### W2: Time Complexity
> The time complexity of our method, given the sparsity of graph Laplacian matrix, is $\mathcal{O}(\kappa E)+ \mathcal{O}(k^2n)$ for the outer loop (Lanczos) and the QR factorization, respectively, where $\kappa \ll n$ is the number of computed eigenvectors and $k \ll n$ is the number of linear constraints.

---

> > ### Comment · Reviewer_8BZ5 · 2023-11-23
> > **Followup on Author Response**
> >
> > Thank the authors for their responses to my questions. However, there are several issues that need more attention:
> >
> > - W1: the introduction of Neumann features is still not well motivated: neither in terms of performance for link prediction and benefits over DRNL and other labeling tricks nor computational efficiency with quantitive analysis compared to other subgraph-based methods.
> >
> > - Q3: I appreciate the authors’ explanation and update on Fig. 1. However, it still seems less intuitive to me how the $k$-hop subgraph is generally encoded in the constraints, the correspondence between nodes and the constraint matrix $C$, and how different constraint functions are designed in principle.
> >
> > - Q4/Q5: Like other subgraph-based models, the proposed method also needs to materialize for each link as Table 2 shows, which brings my major concern over its scalability and time complexity (W2). Even theoretical complexity analysis shows its potential, but it also reveals the size and density of the graph would greatly affect the empirical performance. It would be great to show the wall-clock time comparison with baseline models, especially on large graph benchmarks, say other datasets in OGBL.
> >
> > - Q5.3 / W1: I respectfully disagree with the claim that SEAL ignores the pairwise relation between nodes. DRNL adopted by SEAL encodes each node in the extracted subgraph with the shortest path distance from the queried two nodes, which injects the pairwise relations between nodes in the target link that are the key differentiating it from vanilla GNNs for link prediction. Meanwhile, it is surprising to see the performance gap between LanczosNet and SEAL, especially on collab when both adopt DNRL.
> >
> > I appreciate the authors’ efforts in improving the quality of the submission. However, based on the issues listed above, the manuscript still needs to improve its motivation, clarity, and organization, with further analysis to support the claim. I will stand by my initial rating and encourage the authors to incorporate the feedback and comments from all reviewers.

---

> ### Author Response · Authors · 2023-11-23
>
> We highly appreciate your effort in reviewing our paper and providing good feedback and constructive questions that further improve our paper.
>
> ### W1 & Q5.3.:
> We introduce a novel eigenbasis that addresses the significant limitations of MPNNs. So, our introduction mainly focuses on the limitations of MPNNs in general and how our proposed eigenbasis can address them. We do experiments on LP tasks because their success depends on counting the sub-structures and triangles, which can be resolved by encoding linear constraints into our eigenbasis and discriminating between the k-regular graphs. Also, as discussed in the introduction, the major limitation of previous subgraph-based link prediction is just leveraging oversimplified pairwise node representation features. DRNL, as you mentioned, strictly provides the distance relation "just" between the query nodes that we want to predict the link for and the other individual nodes. No more information is provided by the DRNL, specifically regarding the relation between "other nodes in the graph with each other" or "any set of nodes" (any linear constraints between nodes/edges), which leads to more expressivity (This is what DRNL is missing).
> On the other hand, LLwLC, given encoding "relation between a set of nodes/edges" in the input graph by encoding it as a linear constraint to the eigenbasis, can address this issue with time complexity linear to the number of nodes. (However, I agree with the reviewer that, given SEAL utilizing DRNL, it can provide oversimplified distance node representation. However, as I mentioned, it does not encode the relation between the graph's set of nodes/edges. So, LLwLC has stronger expressivity power than any labeling method by providing this information in its constraint matrix).
> ### Q3:
> We explain our subgraph policy extraction in detail in section 3.2 of our paper. As we explained, we investigate two policies: "Constraints C", where we encode "vertex deleted subgraph", which removes one node for any k-hop subgraphs to make each constraint.   Also, to encode Neumann constraints for k-hop subgraphs,  as we explained that the Neumann eigenvalue constraints encode the boundary conditions of the input graph, for the k-hop subgraph, for its first constraint, we only consider nodes in the boundary (between (k-1)-hop subgraph and k-hop subgraph; which provides us link representation expressivity). The second constraint also encodes the relation between the k-1 hop subgraph. Figure 1 shows the Neumann constraint (an example of encoding link representation)  for two hops away nodes. The main difference between the two graphs is the first constraint, which encodes the relation between purple and green nodes (on the left, two purple nodes (two hops away) are connected to two green nodes (one hop away) (the color corresponds to the constraint on each node (1, 1, -1, -1)), while on the right the boundary nodes are connected to one node in two hops away (so the corresponding constraint is 2, -1, -1). The boundary nodes are different, so the link representation is different. So, the network will be capable of distinguishing between the two graphs).
> ### Q4/Q5:
> We extended our manuscript with a comparison of the computational complexities. A wall clock comparison will be provided in a potential camera ready.

---

### Author Response · Authors · 2023-11-14

# General Comment
We appreciate the time and effort reviewers devoted to the reviewing process. All of the reviewers consider the effectiveness of our novel LLwLC. In the following, we address the comments. At the same time, we notice some misunderstandings of our work. We will clarify these confusions and address the concerns below. We revised the manuscript to address the concerns of all reviewers regarding the presentation of our initial paper. We would appreciate if the reviewers would reconsider their decisions.

---

### Meta-Review · Area_Chair_prxL · 2023-12-08

**Metareview:**

The paper introduces a novel method, LLwLC, which extends LanczosNet for representation learning of induced subgraphs in the context of link prediction. It formulates these subgraphs as Neumann boundary conditions, aiming to enhance the expressiveness of message-passing neural networks (MPNNs). The paper, however, has the following limitations: 1. Insufficient clarity in establishing the connection between the proposed Neumann constraints and subgraphs. 2. Lack of comprehensive experimental evidence, particularly in large-scale datasets and against stronger baselines. 3. The need for clearer organization and presentation, with improved figures, reference formats, and visual representations. In particular, as pointed out by reviewer 8BZ5, the proposed method still needs to materialize a subgraph for each link as Table 2 shows, which brings concerns over its scalability and time complexity. Even theoretical complexity analysis shows its potential, there is no wall-clock time comparison with baseline models. The paper needs a more thorough empirical validation to show its performance/efficiency advantages over state-of-the-art baselines.

**Justification For Why Not Higher Score:**

Claim to be more efficient but no wall-clock time comparison.

**Justification For Why Not Lower Score:**

N/A

---

### Decision · Program_Chairs · 2024-01-16

Reject